# Quantitative Evaluation of a Fully Automated Planning Solution for Prostate-Only and Whole-Pelvic Radiotherapy

**DOI:** 10.3390/cancers16223735

**Published:** 2024-11-05

**Authors:** Jessica Prunaretty, Baris Ungun, Remi Vauclin, Madalina Costea, Norbert Bus, Nikos Paragios, Pascal Fenoglietto

**Affiliations:** 1Institut du Cancer de Montpellier (ICM), 34090 Montpellier, France; pascal.fenoglietto@icm.unicancer.fr; 2TheraPanacea, 75004 Paris, France; b.ungun@therapanacea.eu (B.U.); r.vauclin@therapanacea.eu (R.V.); m.costea@therapanacea.eu (M.C.); n.bus@therapanacea.eu (N.B.); n.paragios@therapanacea.eu (N.P.)

**Keywords:** deep learning, auto-planning, artificial intelligence, prostate cancer, VMAT

## Abstract

In recent years, advanced radiotherapy techniques such as intensity-modulated radiation therapy (IMRT) and simultaneous integrated boost (SIB) irradiation have played a key role in enhancing the precision of radiation delivery to tumors. However, these advancements have also increased the complexity of treatment planning by involving a trial-and-error approach, resulting in greater variability between operators and longer planning times. The automation of planning processes has shown promise in standardizing treatment plans while maintaining their quality and reducing workload. Additionally, deep learning-based fully automated planning solutions have become a significant focus of research in radiation oncology. In this study, we propose a single, end-to-end pipeline for normo-fractionated prostate-only and whole-pelvic cancer treatments, requiring minimal human input and producing a machine-deliverable volumetric modulated arc therapy (VMAT) plan. A comprehensive clinical evaluation was performed, incorporating both dosimetric analysis and plan deliverability assessment.

## 1. Introduction

Prostate cancer (PC) is the second most commonly diagnosed cancer in men and the eighth leading cause of cancer-related deaths worldwide, according to the latest GLOBOCAN study [1]. In recent years, advanced radiotherapy techniques such as intensity-modulated radiation therapy (IMRT) and simultaneous integrated boost irradiation (SIB) have played a crucial role in improving the precision of radiation delivery to tumors. These techniques effectively maximize the target dose while minimizing toxicity to normal tissues and sparing the surrounding organs at risk (OARs). Despite these benefits, the increased complexity of these treatment planning techniques, due to the trial-and-error approach, has led to greater inter-operator variability and longer planning times [2,3].

The strategy of planning automation has shown promising results in standardizing treatment planning while maintaining plan quality and reducing workload [4,5]. Three methods, already commercialized, have demonstrated their performance in overcoming these challenges. First, knowledge-based planning (KBP) uses knowledge from previous cases to predict an achievable dose for a new case in a similar population [6,7]. The multicriteria optimization (MCO) method is based on pareto-optimal plan proposals where one criterion cannot be improved without worsening at least one other criterion [8,9]. Finally, the template-based planning method uses an iterative approach involving progressive optimization that mimics the planning process by a skilled planner. This method requires the creation of a wish list, including beam setup, dose prescriptions, and planning objectives for each treatment, using site-specific clinical data [10,11]. However, the use of any of these algorithms still requires thoughtful input or adjustment from the human planner, such as tuning weights, setting appropriate dose–volume histogram constraints, or identifying the right trade-offs.

Recently, deep learning-based dose distribution prediction using patient anatomy has emerged as a major research focus in radiation oncology. Several studies have investigated the potential of using these predictions to guide inverse planning, specifically dose mimicking, to generate a treatment plan [12,13,14]. In addition, the plan generation directly from dose distribution predictions, bypassing the inverse planning process, has also been explored [15,16,17,18]. In these pioneering studies, dose distribution predictions were used as a crucial intermediate step in the development of a radiotherapy plan based on the patient’s anatomy.

In this study, we propose a single end-to-end pipeline for normo-fractionated prostate-only and whole-pelvic cancer treatments that requires minimal human input and generates a machine-deliverable volumetric modulated arc therapy (VMAT) plan as the output. The aim of this approach is to streamline treatment planning by eliminating the need for expert intervention between contour approval and dosimetric review for plan sign-off.

## 2. Materials and Methods

### 2.1. Patient Data Testing

Twenty patients treated for prostate cancer and twenty patients treated for prostate cancer with pelvic nodes were randomly and retrospectively selected. The patients underwent a 2.5 mm slice thickness computed tomography (CT) scan (GE Optima CT580, General Electric Healthcare, Waukesha, WI, USA) in the supine position.

The prostate was defined as the clinical target volume (referred to as CTV2). The seminal vesicles were delineated in their proximal part, specifically 2 cm from their attachment to the prostate in the cranio-caudal and lateral directions. CTV1 was defined as the junction of CTV2 and the seminal vesicles. In cases where the pelvic nodes were irradiated, CTVn was defined according to the recommendations of Vieillot et al. [19]. The rectum was outlined 2 cm above and below the CTV1. The bladder was contoured on each slice where it appeared. The femoral heads were defined from the acetabular head to the trochanter. The planning target volumes were expanded by 1 cm (0.5 cm posteriorly) from CTV2 and CTV1. In cases of pelvic nodal irradiation, 7 mm was added around CTVn. Detailed patient descriptions are provided in Appendix A.

The study was conducted with a schedule of 80 Gy in 40 fractions to the prostate and simultaneously 56 Gy to the seminal vesicles and pelvic nodes according to the ICRU 83 [20], GETUG-06 [21], and RecoRad [22] recommendations. The dose constraints for the PTVs and organs at risk (OARs) are shown in Table 1.

### 2.2. Manual Treatment Planning

The patients were treated using either the TrueBeam Stx or Ethos linear accelerators, utilizing the volumetric modulated arc therapy (VMAT) technique. The inverse optimization objectives for both the target volumes and the organs at risk were manually defined by planners based on clinical goals. Optimization parameters were iteratively adjusted until the treatment plan met clinical standards. Dose calculations were performed using the AAA (v15.6, Varian) or AcurosXB Ethos (v1.1, Varian) algorithms. The number of arcs and collimator angles were determined by the operator, considering the anatomical complexity of each patient. Further details of the treatment protocols are provided in Appendix A.

### 2.3. Automated Treatment Planning

A treatment planning pipeline was developed that takes as its input a planning CT with organs-at-risk (OARs) and planning target volume (PTV) contours, the targeted linac machine, and the prescription dose. A cohort of 324 patients that included prostate cases was retrospectively collected, following a prescription of 80 Gy to the prostate and, simultaneously, 56 Gy to either (a) the seminal vesicles or (b) the seminal vesicles and pelvic lymph nodes, delivered in 40 fractions. Individual cases were examined to filter out cases with prostheses or any outliers (e.g., major anatomical abnormalities such as large bladder diverticulitis or large inguinal hernias). For each case, a lookup table was built matching the structure names present in the RTSTRUCT DICOM file to a set of standardized names for targets and OARs, as pertinent to the indication. The patient age information was used to perform an age-stratified random split of the data to ensure that the test set’s distribution of ages matched that of the patient population undergoing RT for prostate cancers.

The planning CT, OARs, and PTVs were resampled to the dose map resolution (2.5 mm), and the PTV contours were filled with their prescription to better condition the network. These volumes were then concatenated and cropped randomly into patches.

The first step was to generate a dose prediction. For that, a convolutional neural network (CNN) architecture was designed using a 3D U-Net with skip connections [23]. The network was trained with a combination of a reconstruction loss and an adversarial loss. The reconstruction loss is a weighted mean absolute error (MAE). The PTV and OAR voxels were given a greater weight to promote better PTV coverage and OAR sparing. An adversarial loss was added by training a discriminator CNN to differentiate between the real and predicted dose maps [24]. It resulted in more conformal dose maps with steeper gradients. The deep learning model was trained on 238 cases, and a held-out set of 86 cases was used for model validation.

The second step was to generate a direct aperture VMAT plan optimization that sought to mimic the predicted dose. Custom implementations of two types of gradient-based optimization algorithms were used. The optimization task employed loss functions that penalized, for each anatomical or planning structure, deviations from the voxel-wise doses predicted by the deep learning model. Objective weights were selected via hyperparameter tuning.

The training of the deep neural network was conducted over 3100 epochs, for a total of 738,000 training steps, using the SiLU activation function [25] and the Adam optimizer [26], with a learning rate of 4.5 × 10^−6^. The training was performed on a single 1 GPU, with a training time of several days.

The output treatment plan was a 2-full-arc VMAT plan using the TrueBeam Stx linear accelerator. Once automatically generated, the plans were imported into Eclipse TPS and normalized on the PTV2, depending on the coverage of the target volumes and the doses to the organs at risk initially obtained. The normalization was either 100% of the dose covering 50% of the volume or 95% of the dose covering 95% of the volume.

### 2.4. Plan Comparison

All the treatment plans were transferred to the Eclipse treatment planning system for the purpose of conducting a side-by-side comparison. Dose metrics were compared using the dose constraints provided in Table 1. The target volume coverage was assessed using the doses received at 98% and 95% of the volume (D_98%_, D_95%_) for the PTVs, respectively. For the bladder and the rectum, the maximal and median doses (D_2%_, D_50%_) were calculated. Regarding the femoral heads, the dose received by 0.1% and 5% of the volume (D_0.1%_, D_50%_) were recorded. In the case of whole-pelvic irradiation, the bowel volume receiving 40 Gy and 30 Gy (V_40Gy_, V_30Gy_) was reported.

The homogeneity index (HI) within the PTVs was evaluated using the following formula:$$HI=\frac{D_{2\%}-D_{98\%}}{D_{50\%}}$$
where D_2%_, D_98%_, and D_50%_ are the doses received by 2%, 98%, and 50% of the PTV, respectively [20].

The dose conformity was assessed using the conformity index (CI), which is defined as:$$CI=\frac{{\left(V_{95\%\left(PTV\right)}\right)}^{2}}{V_{PTV}\times V_{95\%(Body)}}$$
where V_95%_ (PTV) and V_95%_ (Body) are the volumes receiving at least 95% of the prescribed dose for the whole PTV and body, respectively [27]. VPTV is the volume of the PTV.

The Wilcoxon signed rank test was used to determine the significant difference between the manual and automated plan metrics. A Bonferroni correction was applied, and the significance level was set at 0.002.

### 2.5. Quality Assurance (QA)

Pre-treatment QA was performed for each treatment plan using portal dosimetry results. The assessment metric was the global gamma pass rate with a 3%3 mm and 2%2 mm criteria and a 10% threshold. Moreover, the modulation complexity score applied to VMAT treatment (MCSv) and defined by Masi et al. [28] was computed. The MCSv score ranges from 1 for a simple unmodulated field and decreases towards 0 with increasing inherent plan complexity. Finally, the total number of monitor units (MUs) was reported for each plan.

## 3. Results

This study evaluated the viability of employing a fully automated planning solution for prostate-only and whole-pelvic radiotherapy. The generation times for the automatic plans for prostate-only and prostate-with-lymph-nodes cases are displayed in Table 2.

Figure 1 and Figure 2 show an example of dose distributions comparing automated and manual treatment plans for prostate-only localization. Another example for whole-pelvic irradiation is shown in Appendix A.

Table 3 shows the percentage of treatment plans that did not comply with the dose constraints with the range of the out-of-tolerance values [Min–Max]. Only the D_95%_ of PTV2 and the maximum doses (D_2%_) for the rectum and bladder were affected. A greater number of automated plans did not meet the dose constraints for prostate-only localization, while a greater number of manual plans were affected for the whole-pelvic region. However, the deviations were less than 1 Gy and 0.4 Gy for the manual and automated plans, respectively.

Table 4 and Table 5 provide a summary of the dose metric results for the planning target volumes (PTVs) and organs at risk (OARs) in both the automated and manual treatment plans for the prostate and pelvic regions, respectively. Regarding the prostate localization (Table 3), PTV2 coverage was slightly improved with the manual plans while PTV1 coverage and homogeneity were increased with the automated plans. The dose received by 50% of the rectum volume was significantly decreased with the automated plans. The conformity indices were not statistically different between the two plans.

Regarding the whole-pelvic localization (Table 4), the PTV coverage was slightly increased with the manual plans. The absolute difference was lower than 1 Gy. The OAR protection was not significantly different between the plans.

In terms of treatment delivery, the automated plans resulted in an increase in MUs of 26.8% and 7.9% for the prostate-only and whole-pelvic localizations, respectively, compared to the manual plans (Table 6). Pre-treatment verification showed that the median gamma pass rates remained above 99% and 96% for the 3%/3 mm and 2%/2 mm criteria, respectively, regardless of plan type. In addition, no difference in modulation complexity scores was observed. Finally, no statistically significant correlations were found between any of the gamma criteria tested and the number of MUs or MCSv for each arc.

## 4. Discussion

In this study, we presented a single 3D U-Net coupled with a VMAT direct aperture optimization that was able to directly generate deliverable VMAT treatment plans for prostate-only and whole-pelvic cancer. To our knowledge, this is the first study to present and comprehensively evaluate a single deep learning-based, fully automated treatment planning pipeline without the need to convert model predictions into deliverable VMAT treatment plans for these both localizations. Heileman et al. [29] presented one of the first deep learning-based models capable of generating deliverable DICOM RT treatment plans that could be directly executed by a linear accelerator for prostate-only VMAT radiotherapy. The model utilizes an encoder–decoder network and is designed to predict multileaf collimator (MLC) motion sequences. Although the method was promising, the quality of the resulting plans was not good enough when compared to the manual plans. For example, Lempart et al. [18] presented a 2.5D U-Net model trained on a combination of three consecutive image slices and corresponding segmentations showing suitable clinical and deliverable plans for prostate cancer. However, their workflow between the dose prediction and the TPS was not fully automated, and some manual interventions were needed. McIntosh et al. [30] conducted the first study to generate IMRT plans based on a collapsed cone convolution dose engine from knowledge-based per-voxel dose prediction using machine learning for head and neck cancer. Despite improved dosimetric results compared with the clinical plans, the plan delivery part was not investigated.

The automated plans provided adequate treatment plans (or minor deviations) with respect to the dose constraints, and the quality of the plans was similar to the manual plans. When certain metrics fell outside the tolerance range, the absolute dose difference was less than 0.4 Gy. It is reasonable to assume that these automated plans would be clinically acceptable, given that some manual plans were approved by physicians with an absolute dose difference of up to 1 Gy. However, medical approval is still required to confirm this assumption.

In terms of planning and treatment efficiency, automated treatment planning resulted in an increase in MUs. This suggests an increase in plan complexity, but without any impact on the quality assurance. Nevertheless, no impact was found on the plan complexity and the quality assurance.

Although the time savings have not been fully quantified, it is reasonable to assume that the plan preparation time will be reduced. In our clinical experience, the time required for a planner to manually generate a clinically acceptable prostate VMAT plan ranged from 30 to 90 min, depending on anatomical complexity. The generation of the automated plans took less than 10 min without intervention. In fact, manual plans typically require several optimization trials before reaching a clinically acceptable result, whereas the automated plan is a ‘one-click’ output. The automated plan generation process, in fact, assumes that no manual adjustments will be incorporated. Our proposed deep learning-based approach demonstrates significant potential and could enhance the overall treatment planning workflow for prostate cancer patients by speeding up the process.

The main limitation of the study was the use of a single-dose prescription. For example, hypofractionated schemes require separate training sets and models due to different dose constraints on the organs at risk. Future research should focus on making the automated treatment planning process more universally applicable.

## 5. Conclusions

This study showed the feasibility of a deep learning-based fully automated treatment planning pipeline that generates deliverable high-quality plans that are comparable with manually made, clinically approved plans in terms of dosimetry and machine deliverability. In the future, this framework should be extended to other prescriptions and anatomical regions.

## Figures and Tables

**Figure 1 cancers-16-03735-f001:**
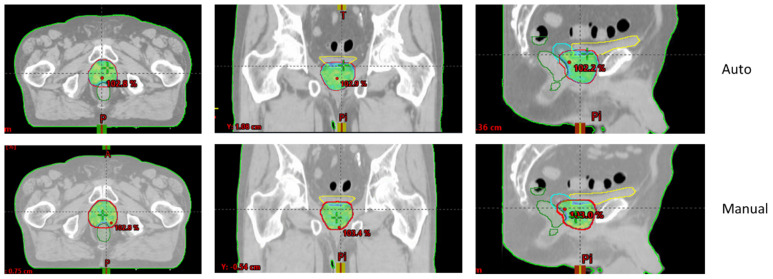
Example of 76 Gy isodose for automated (upper—“Auto”) and manual (lower) treatment plans for prostate-only localization (Patient n° 7).

**Figure 2 cancers-16-03735-f002:**
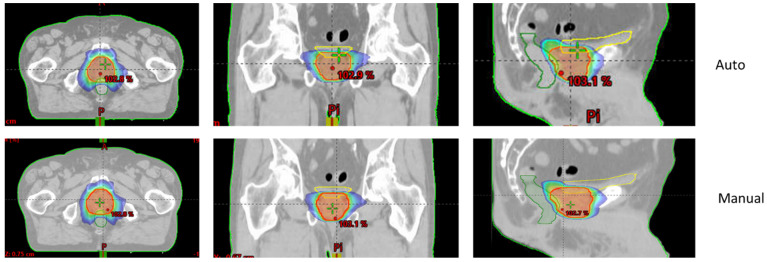
Example of 40 Gy isodose for automated (upper—“Auto”) and manual (lower) treatment plans for prostate-only localization (Patient n° 7).

**Table 1 cancers-16-03735-t001:** Planning target volume (PTV) and organs-at-risk dose constraints.

PTV	D_98%_ > 90%
	D_95%_ > 95%
	D_50%_ = 100%
	D_2%_ ≤ 107%
Rectum	D_2%_ < 76 Gy
	D_50%_ < 60 Gy
Bladder	D_2%_ < 80 Gy
	D_50%_ < 70 Gy
Femoral head	D_0.1%_ < 55 Gy
	D_5%_ < 50 Gy
Bowel	V_40Gy_ < 200 cc
	V_30Gy_ < 450 cc

**Table 2 cancers-16-03735-t002:** Generation time for the prostate-only and whole-pelvic plans on a server equipped with 4 NVidia 3070 and 1 GPU, and the same server with 4 GPUs.

	4 NVidia 3070	
	1 GPU	4 GPUs
Prostate-only	9.8 min	4.4 min
Whole-pelvic	35.6 min	13.4 min

**Table 3 cancers-16-03735-t003:** Percentage of treatment plans that did not meet the dose constraints with the range of the out-of-tolerance values [Min–Max].

	PTV2		Rectum		Bladder	
	D_95%_ > 76 Gy		D_2%_ < 76 Gy		D_2%_ < 80 Gy	
	Manual	Auto	Manual	Auto	Manual	Auto
Prostate-only	5% [75.84 Gy]	15% [75.85; 75.98 Gy]	0%	0%	5% [80.14 Gy]	20% [80.05; 80.34 Gy]
Whole-pelvic	5% [75.84 Gy]	0%	15% [76.14; 76.22 Gy]	5% [76.05 Gy]	20% [80.07; 80.92 Gy]	0%

**Table 4 cancers-16-03735-t004:** Dose metric, homogeneity (HI), and conformity index (CI) results (mean ± standard deviation) for the planning target volumes (PTV1 and PTV2) and organs at risk (OARs) for the automated (“Auto”) and manual treatment plans for prostate-only localization. Values are in bold where *p* < 0.002.

		Auto	Manual	*p*-Value
PTV2	**D_98%_ (Gy)**	**74.12 ± 0.33**	**74.7 ± 0.53**	**0.0003**
	D_95%_ (Gy)	75.99 ± 0.05	76.33 ± 0.47	0.006
	D_50%_ (Gy)	80.71 ± 0.57	80.20 ± 0.51	0.003
	D_2%_ (Gy)	82.48 ± 0.61	82.17 ± 0.84	0.100
	HI	0.10 ± 0.01	0.09 ± 0.01	0.002
	CI	0.90 ± 0.02	0.90 ± 0.02	0.705
PTV1	**D_98%_ (Gy)**	**58.53 ± 1.71**	**56.55 ± 1.78**	**3.63 × 10^−5^**
	**D_95%_ (Gy)**	**60.97 ± 2.69**	**58.96 ± 2.95**	**4.77 × 10^−5^**
	D_50%_ (Gy)	79.76 ± 0.79	79.47 ± 0.62	0.050
	D_2%_ (Gy)	82.38 ± 0.62	82.07 ± 0.83	0.097
	**HI**	**0.30 ± 0.02**	**0.32 ± 0.02**	**0.0003**
	CI	0.61 ± 0.02	0.61 ± 0.04	0.793
Rectum	**D_50%_ (Gy)**	**22.60 ± 3.60**	**25.22 ± 4.01**	**0.0001**
	D_mean_ (Gy)	29.88 ± 3.32	31.36 ± 3.71	0.003
	D_2%_ (Gy)	74.47 ± 1.10	74.74 ± 0.48	0.247
Bladder	D_50%_ (Gy)	36.14 ± 13.71	36.06 ± 13.65	1
	D_mean_ (Gy)	40.55 ± 8.75	40.20 ± 8.55	0.452
	D_2%_ (Gy)	79.36 ± 0.63	79.26 ± 0.42	0.4
Femoral Head	D_0.1%_ (Gy)	39.40 ± 5.40	38.18 ± 8.31	0.674
	D_5%_ (Gy)	32.48 ± 4.44	31.35 ± 7.31	0.261

**Table 5 cancers-16-03735-t005:** Dose metric, homogeneity (HI), and conformity index (CI) results (mean ± standard deviation) for PTVs and organs at risk (OAR) for the automated (“Auto”) and manual treatment plans for whole-pelvic localization. Values are in bold where *p* < 0.002.

		Auto	Manual	*p*-Value
PTV2	D_98%_ (Gy)	74.50 ± 0.33	74.90 ± 0.78	0.017
	**D_95%_ (Gy)**	**76.09 ± 0.17**	**76.72 ± 0.63**	**0.0004**
	D_50%_ (Gy)	80.21 ± 0.25	80.31 ± 0.58	0.605
	D_2%_ (Gy)	82.06 ± 0.51	81.99 ± 0.72	0.765
	HI	0.10 ± 0.01	0.10 ± 0.01	0.716
	CI	0.90 ± 0.02	0.88 ± 0.02	0.023
PTV1	**D_98%_ (Gy)**	**53.02 ± 0.69**	**53.97 ± 1.15**	**0.0013**
	D_95%_ (Gy)	54.48 ± 0.60	54.90 ± 0.96	0.153
	D_50%_ (Gy)	58.06 ± 0.58	57.71 ± 0.83	0.131
	D_2%_ (Gy)	81.33 ± 0.50	81.32 ± 0.71	0.701
	HI	0.49 ± 0.02	0.47 ± 0.02	0.015
	CI	0.78 ± 0.02	0.78 ± 0.02	0.248
Rectum	D_50%_ (Gy)	27.33 ± 4.81	29.42 ± 4.40	0.179
	D_mean_ (Gy)	33.47 ± 3.51	35.05 ± 3.68	0.114
	D_2%_ (Gy)	74.60 ± 1.13	74.55 ± 1.24	0.627
Bladder	D_50%_ (Gy)	50.84 ± 10.86	49.60 ± 11.40	0.165
	D_mean_ (Gy)	50.33 ± 7.05	49.35 ± 7.30	0.202
	D_2%_	79.16 ± 0.41	79.14 ± 0.81	0.409
Bowel	V_30Gy_ (cc)	281.62 ± 86.34	300.64 ± 90.01	0.216
	V_40Gy_ (cc)	151.74 ± 49.14	164.94 ± 49.33	0.070
Femoral Head	D_0.1%_ (Gy)	45.80 ± 3.81	46.52 ± 4.43	0.330
	D_5%_ (Gy)	36.31 ± 3.19	37.57 ± 4.33	0.019

**Table 6 cancers-16-03735-t006:** Median [Min–Max] values of total monitor units (MUs), modulation complexity score (MCSv), and gamma passing rates (3%3 mm and 2%2 mm criteria) for manual and automated (“Auto”) treatment plans.

	MU		MCSv		Gamma Pass Rates			
					3%/3 mm		2%2 mm	
	Manual	Auto	Manual	Auto	Manual	Auto	Manual	Auto
Prostate-only	549.5 [495;724]	696.8 [599.6;805.6]	0.176 [0.103;0.249]	0.173 [0.147;0.196]	100 [98.6;100]	99.3 [96.9;100]	99.8 [88.4;100]	96.6 [92.7;98.5]
Whole-pelvic	638 [545;945]	688.7 [624.2;747]	0.174 [0.145;0.226]	0.153 [0.122;0.175]	99.8 [95;100]	99.5 [97.5;99.8]	98.6 [93.4;100]	97.3 [94;99.5]

## Data Availability

The datasets used and/or analyzed during the current study are available from the corresponding author on reasonable request.

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
