# Peer review of "Quantitative Evaluation of a Fully Automated Planning Solution for Prostate-Only and Whole-Pelvic Radiotherapy"

_cancers, 2024, doi:10.3390/cancers16223735_

Round 1

Reviewer 1 Report

Comments and Suggestions for Authors

Thank you for your submission.

In the introduction, please further highlight the need and importance for full automation in prostate planning.

In the method, many technical details are missing. Please include all relevant details, at least in supplementary material, for the deep learning model used, training parameters (learning rate, activation function, epochs, backbone architecture, convolutional layers, etc) and image pre- and post-processing parameters. It is of vital importance to display these settings to facilitate knowledge synthesis and result reproducibility. If the software used is proprietary, you are highly suggested to seek consent from the vendor to disclose it as appropriate. Make sure they are presented systematically, especially in section 2.3.

In the results, the training and testing outcomes should be displayed in table format, especially those involving many numbers and parameters with various units. This will aid readers in interpreting it much faster and more accurately. Table 3 has major flaws in "." and "," consistency. Table 4 has units missing. All full forms for abbreviation are missing for most tables throughout the manuscript.

In the discussion, please enlighten the reader about how manual adjustment can be incorporated into the plan generation process. "Human in the loop" is important for any medical automation tools to ensure transparency and explainability.

Please provide adjustments for the above. This work has the potential to facilitate personalised care in prostate radiotherapy.

Author Response

Dear reviewer,

We sincerely thank the reviewer for their insightful comments and valuable suggestions, which have helped us improve the clarity and quality of our manuscript. You will find in attached file the responses of your comments.

Best regards,

Reviewer 2 Report

Comments and Suggestions for Authors

the research topic is fascinating and could improve understanding of new methods for medication of prostate cancers. the introduction is very explanatory and the authors emphasize the fact that prostate cancer is one of the most abundant diseases of men over 50 years old. the authors briefly analyzed already implemented techniques summarizing their advantages and disadvantages and pointing out their influence on surrounding organs. The rest of the introduction explains the study plan, gives the necessary information about deep learning dose techniques, and compares similar studies.

the materials and methods describe the precisely all-important parts that are needed for successfully repeating the experiments and the reader can easily repeat it.

the results are presented appropriately, focusing on the study's aim. the discussion thoroughly analyzed the presented results pointing out all the advantages and flaws of the presented methods, the conclusion is precise, drawn from the results, and points out the benefits of the proposed method. 

 the number of figures and tables is adequate, improves paper quality, and allows the reader to make conclusions on their own.

but I have few suggestions:

first better explanation of why the authors chose the median value and if there is the possibility to compare the median value with average values of total MUs.

second analysis of whether there are statistically significant differences between manual and auto treatment plans.

third how the database for data learning was built and the criteria for selection of it..

Author Response

Dear reviewer,

We sincerely appreciate your insightful comments and valuable suggestions, which have greatly enhanced the clarity and quality of our manuscript. Please find in attached file the responses to your comments.

Best regards,
